# SEMANTIC MEMORY GUIDED DIFFUSION NETWORKS FOR IMAGE-TO-LONG TEXT GENERATION

## ABSTRACT

Automatic describing image with comprehensive textual content is often demanded by different real-world applications, which motivates image-to-text generation tasks such as image captioning. However, conventional tasks mainly focus on generating short text, which often fail to deal with challenging scenarios that long text is inevitable required to describe enriched and diversified visual contents. Therefore, a more generic solution, which should be able to generate text with arbitrary length (long text in most cases), is expected to overcome limitations from existing approaches such as inability to generate sufficiently comprehensive and complete textual content and ensure semantic coherence in it. To address such limitations, we propose a dedicated solution, semantic memory guided diffusion networks (SeMDiff), for image-to-long text generation (I2LTG), which explicitly captures salient semantics from the visual contents, and further process and enhance them by memory networks to facilitate the text generation process. Specifically, we employ semantic concepts as the vehicle to deliver and process semantics embedded in images, where they are predicted from each image and matched with memory vectors and serve as the condition to guide diffusion networks for iterative generation. Experimental results on three public datasets and a new proposed one with more than 54K instances demonstrate the superiority of our approach compared to previous state-of-the-art solutions. Further analyses illustrate that our approach offers an effective diffusion-based solution with external guidance for long text generation under different cross-modal settings. [1]

## 1 INTRODUCTION

Generating image descriptions is one of the most widely applied techniques in artificial intelligence, especially when visual contents are enriched and diversified so that one needs an effective process to produce and organize descriptive texts that cover all semantics in the scenery. To emulate the process, some task such as image captioning (IC) has been developed to do so and achieves promising results (Mao et al., 2015; Rennie et al., 2017; Anderson et al., 2018; Pantazopoulos et al., 2022). However, IC mainly deals with short texts, which often fail to satisfy the demands of challenging scenarios, especially in particular domains where an entire report is expected to be generated with given image, i.e., radiology report generation (RRG) (Jing et al., 2018; Li et al., 2018; Johnson et al., 2019; Liu et al., 2021b; Huang et al., 2023). Therefore, the ability of generating comprehensive long text for images is expected to upgrade existing image-to-text generation approaches.

In performing current image description tasks, existing approaches adopt either AR (Herdade et al., 2019; Huang et al., 2019; Cornia et al., 2020; Hu et al., 2022; Li et al., 2023; Zhu et al., 2023; Liu et al., 2023) (e.g., Transformer (Vaswani et al., 2017)) or non-AR models (Lee et al., 2018; Gao et al., 2019a; Guo et al., 2020; Zhou et al., 2021) as their foundation architecture, by predicting words in a sequence or producing all words in parallel, respectively. In forcing these approaches to generate long texts, they all have difficulties in producing semantically coherent texts with both AR and non-AR manner. Particularly, AR solutions are susceptible to error propagation if incorrect contents are half-way generated, so that contextually irrelevant contents are always observed accordingly, thus exacerbate the coherent problem. Although some RRG studies (Chen et al., 2020; 2021; Qin & Song, 2022; Tanida et al., 2023; Omkar Thawkar & Khan, 2023; Tu et al., 2023) extend AR solutions with task- and domain-specific heuristics, they cannot guarantee comprehensive and coherent content

---

[1]Code, models, and the proposed dataset will be open-sourced in the final version of this paper.

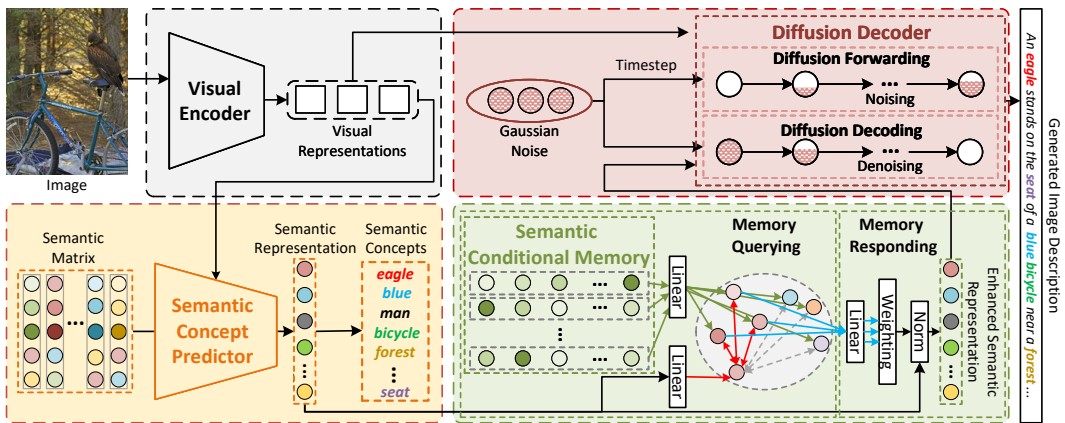

Figure 1: The overview architecture of our approach for I2LTG, which consists of four components, namely, the visual encoder, the semantic concept predictor, the semantic conditional memory, and the diffusion decoder, which are represented in grey, yellow, green, and red background, respectively. An example input image with its output text is provided for better demonstration.

generation. Therefore, in terms of generation mechanism, non-AR approaches are relatively optimal than AR ones to avoid sequential error propagation. However, they are verified only on short text generation task in most cases, e.g., IC, and it is unclear if being applied to long text generation especially they also have their own limitations such as word repetition issue (Luo et al., 2022). As a result, to explore effective long text generation with non-AR approaches, it is valuable to carefully design guidance and enhancement that adapt to this task.

In this paper, we propose semantic memory guided diffusion networks (SeMDiff) for image-to-long text generation (I2LTG) with three main components, namely, semantic concept predictor (SCP), semantic conditional memory (SCM), and diffusion decoder (DD). In our approach, we adopt semantic concepts as intermediate media to transport essential semantic information in image to text generation process, where they are captured from image by SCP and enhanced in SCM, then serve as the guidance for DD to iteratively generate final texts. Specifically, SCM is the distinctive design in this work that enhances the representation of semantic concepts with specific image-text correlation information stored in its most related memory vectors so as to provide precise control that piloting diffusion networks in generating comprehensive and coherent long texts. We evaluate our approach on three public datasets, i.e., MIMIC-CXR, CC-SBU and Localized Narratives (LN) and a new proposed one designed for I2LTG in this work, namely, COCO-Long Text (COCO-LT). Experimental results on them illustrate the superiority of our approach against state-of-the-art counterparts under different image description generation settings. Further analysis on different components of our approach illustrates that the SCP provides strong guidance for iterative refinement of DD, which allows the model to perform a more organized generation process, with SCM further ensuring the preciseness of the guidance for each iteration, guaranteeing the resulted semantic coherent texts.

## 2 THE APPROACH

Given an input image $\mathcal{I}$, our approach attempts to generate its description $\widehat{\mathcal{Y}}$ in long text. Figure 1 illustrates the overall pipeline of our approach, which consists of four components, i.e., the visual encoder, the semantic concept predictor (SCP), the semantic conditional memory (SCM), and the diffusion decoder (DD). Specifically, the visual encoder $f_{ve}$ processes the input image $\mathcal{I}$ into visual representations $\mathbf{h}^v$, and SCP $f_{scp}$ predicts semantic concepts $\widehat{\mathcal{S}}$ from a semantic matrix $\mathbf{S}$ that stores the vectors of all possible concepts according to $\mathbf{h}^v$. The SCM $f_{scm}$ further enhances the representations $\mathbf{h}^s$ of $\widehat{\mathcal{S}}$ by matching top-$\mathcal{K}$ memory vectors, resulting in a subset $\widehat{\mathbf{h}}^s$ of $\mathbf{h}^s$. Finally, the DD $f_{dd}$ generates $\widehat{\mathcal{Y}}$ along with $\widehat{\mathbf{h}}^s$ and $\mathbf{h}^s$, where the overall process is formulated by

$$\widehat{\mathcal{Y}} = f_{dd}(f_{ve}(\mathcal{I}), f_{scm}(f_{scp}(f_{ve}(\mathcal{I}), \mathbf{S}), \mathcal{K})) \tag{1}$$

In training, the model is optimized based on the cross-entropy loss $\mathcal{L}_{SCP}$. The final loss $\mathcal{L}$ for the entire approach is then combined with $\mathcal{L}_{SCP}$ and the loss function $\mathcal{L}_{DD}$ of the DD through

$$\mathcal{L} = \beta_1 \mathcal{L}_{SCP} + \beta_2 \mathcal{L}_{DD} \tag{2}$$

where $\beta_1$ and $\beta_2$ are hyper-parameters balancing contributions of $\mathcal{L}_{SCP}$ and $\mathcal{L}_{DD}$, respectively. Following texts present aforementioned components in details according to our pipeline sequence.

## 2.1 THE VISUAL ENCODER

The visual encoder consists of two components, a visual feature extractor $f_{ve}$ and a Transformer-based encoder $f_{te}$, where $f_{ve}$ is a pre-trained vision backbone model (i.e., ResNet-101 (He et al., 2016)). For feature extraction from $\mathcal{I}$, we firstly decompose $\mathcal{I}$ into a series of patches $\{\mathcal{I}_1 \ldots \mathcal{I}_{\mathcal{N}_v}\}$ with $\mathcal{N}_v$ denoting the number of patches, and then adopt the output matrices $[\mathbf{X}_1 \ldots \mathbf{X}_{\mathcal{N}_v}]$ from the last convolutional layer of $f_{ve}$ to feed into $f_{te}$. Finally, $f_{te}$ encodes $[\mathbf{X}_1 \ldots \mathbf{X}_{\mathcal{N}_v}]$ into visual representations $\mathbf{h}^v$, with the overall process formulated by

$$\mathbf{h}^v = f_{te}(f_{ve}(\mathcal{I}_1 \ldots \mathcal{I}_{\mathcal{N}_v})) \tag{3}$$

## 2.2 THE SEMANTIC CONCEPT PREDICTOR

When generating long texts directly with the latent representations extracted from an image, there is potential deficiency that such representations have ambiguities in conveying all essential semantics, so that incoherent or even incomplete image descriptions are generated. To address such ambiguity issue, we propose SCP to explicitly predict semantic concepts, so as to provide accurate supplementary guidance for image representations. Starting from the randomly initialized matrix $\mathbf{S}$ containing a series of semantic vectors $\{\mathbf{s}_1 \ldots \mathbf{s}_{\mathcal{N}_s}\}$ that cover all the possible concepts, we use $f_{scp}$, a transformer based ranker, to predict $\widehat{\mathcal{S}} = \{\widehat{s}_1 \ldots \widehat{s}_{\mathcal{N}_c}\}$ with $\mathcal{N}_c$ concepts (i.e., words in some cases) according to $\mathbf{h}^v$ from the visual encoder, with the process formulated by

$$\widehat{S} = f_{scp}(\mathbf{h}^v, \mathbf{s}_1 \ldots \mathbf{s}_{\mathcal{N}_s}) \tag{4}$$

where the representation $\mathbf{h}_n^s$ of the $n$-th concept $\widehat{s}_n$ is extracted from the last layer of $f_{scp}$ by

$$\mathbf{h}_n^s = f_{scp}(\mathbf{h}^v, \mathbf{s}_1 \ldots \mathbf{s}_{\mathcal{N}_s}; \widehat{s}_1 \ldots \widehat{s}_n) \tag{5}$$

Later we compute the mean pooling of all $\mathbf{h}_n^s$ and use the resulting vector $\mathbf{h}^s$ to represent $\widehat{\mathcal{S}}$. In training, we compute the cross-entropy loss $\mathcal{L}_{SCP}$ between $\widehat{\mathcal{S}}$ and the annotated semantic concepts $\mathcal{S}^*$ in the gold standard image description $\mathcal{Y}^*$. In doing so, we map $\mathbf{h}_n^s$ to a distribution over $\mathcal{V}^s$ with $p_{n,i}^s$ for the probability of the $i$-th concept $v_i$, and choose the concept $\widehat{s}_n$ with the highest probability as output. Then, we compare $\widehat{s}_n$ with the gold standard $y_n^*$ to compute the cross-entropy loss by

$$\mathcal{L}_{S,n} = -\sum_{v_i \in \mathcal{V}^s} p_{v_i}^* \log p_{n,i}^s \tag{6}$$

where $p_{v_i}^*$ is the probability distribution of the gold standard over $\mathcal{V}^s$ with $p_{v_i}^* = 1$ if $v_i = y^*$ and $p_{v_i}^* = 0$ otherwise. Finally, we sum $\mathcal{L}_{S,n}^n$ over all concepts in $\widehat{\mathcal{S}}$ and obtain $\mathcal{L}_{SCP} = \sum_{n=1}^{\mathcal{N}_c} \mathcal{L}_{S,n}^n$.

## 2.3 THE SEMANTIC CONDITIONAL MEMORY

In our approach, we utilize the SCM to enhance the representations of the produced concepts from the SCP with the memory that stores the information in aligning images and texts, so as to provide more precise guidance for the next text generation process. In doing so, SCM is built upon a memory matrix $\mathbf{M}$, which stores a series of $d$-dimension memory vectors $\{\mathbf{m}_1 \ldots \mathbf{m}_{\mathcal{N}_m}\}$ that interact with $\mathbf{h}^s$, with $\mathcal{N}_m$ denoting the number of these vectors. Two main steps are involved in SCM, namely, memory querying (MQ) and memory responding (MR), respectively.

**Memory Querying** In this process, we project $\mathbf{h}^s$ and $\mathbf{m}_i \in \{\mathbf{m}_1 \ldots \mathbf{m}_{\mathcal{N}_m}\}$ into $\mathbf{q}^s$ and $\mathbf{k}_i$ to the same semantic space through two linear transformation matrices $\mathbf{W}_q$ and $\mathbf{W}_k$, respectively, through

$$\mathbf{q}^s = \mathbf{h}^s \cdot \mathbf{W}_q, \quad \mathbf{k}_i = \mathbf{m}_i \cdot \mathbf{W}_k \tag{7}$$

where we use two one-layer perceptrons to model $\mathbf{W}_q$ and $\mathbf{W}_k$, respectively. Then, we compute the latent distance $\mathcal{D}_i$ between $\mathbf{q}^s$ and $\mathbf{k}_i$ by

$$\mathcal{D}_i = \frac{\mathbf{q}^s \cdot \mathbf{k}_i^\mathsf{T}}{\sqrt{d}} \tag{8}$$

Subsequently with $\mathcal{D}_i$, we retrieve the top-$\mathcal{K}$ memory vectors $\{\mathbf{k}_1 \ldots \mathbf{k}_{\mathcal{K}}\}$ from $\mathbf{M}$ and calculate the corresponding importance weight $\omega_i$ for each $\mathbf{k}_i$ by normalization over $\mathcal{D}_i$:

$$\omega_i = \frac{\exp(\mathcal{D}_i)}{\sum_{j=1}^{\mathcal{K}} \exp(\mathcal{D}_j)} \tag{9}$$

**Memory Responding** MR obtains a responded vector $\mathbf{r}$ based on $\{\mathbf{k}_1 \dots \mathbf{k}_{\mathcal{K}}\}$ and their weights $\{\omega_1 \dots \omega_{\mathcal{K}}\}$, and enhance $\mathbf{h}^s$ with the resulted $\mathbf{r}$. In doing so, we project $\mathbf{k}_i$ to the same semantic space of $\mathbf{h}^s$ through a linear transformation matrix $\mathbf{W}_v$, resulting $\mathbf{v}_i$ through

$$\mathbf{v}_i = \mathbf{k}_i \cdot \mathbf{W}_v \tag{10}$$

where $\mathbf{W}_v$ is performed by a one-layer perceptron. Then, we obtain the responded vector $\mathbf{r}$ by

$$\mathbf{r} = \sum_{i=1}^{\mathcal{K}} \omega_i \cdot \mathbf{v}_i \tag{11}$$

Finally, we add $\mathbf{r}$ to $\mathbf{h}^s$ and normalizing ($Norm$) it as $\widehat{\mathbf{h}}^s = Norm(\mathbf{h}^s + \mathbf{r})$, and send $\widehat{\mathbf{h}}^s$ to DD to guide the generation process.

## 2.4 THE DIFFUSION DECODER

The DD ($f_{dd}$) aims to generate $\widehat{\mathcal{Y}}$ based on $\mathbf{h}^v$ and $\widehat{\mathbf{h}}^s$. In doing so, DD performs diffusion forwarding and decoding processes, where forwarding allows DD to learn the ability of reconstructing noisy representation and insert them into final result, so that DD is able to generate $\widehat{\mathcal{Y}}$ through iteratively denoising during the decoding process. Details of these processes are illustrated in following texts.

**Diffusion Forwarding** Given the step $t \sim U(0, T)$ with $T$ denoting the total number of steps, diffusion forwarding firstly adds Gaussian noise $\mathbf{n}$ into the representation $\mathbf{h}_0$ of $\mathcal{Y}^*$, resulting in the noisy representations $\mathbf{h}_t$ at $t$-step. We follow Bit Diffusion (BD) (Chen et al., 2023) to convert tokens in $\mathcal{Y}^*$ into their bit representation ($\mathbf{h}_0$) and compute the representation $\mathbf{h}_t$ at the $t$-th step by

$$\mathbf{h}_t = \sqrt{\bar{\alpha}_t} \cdot \mathbf{h}_0 + \sqrt{1 - \bar{\alpha}_t} \cdot \mathbf{n} \tag{12}$$

Herein, $\bar{\alpha}_t$ is a blending scalar correlated to the noise scheduling strategy of denoising diffusion probabilistic model (DDPM) (Ho et al., 2020), and we use the cosine noising schedule of DDPM. Then, $f_{dd}$ reconstructs $\mathbf{h}_t$ to $\mathbf{h}_0$ based on $\mathbf{h}^v$ and $\widehat{\mathbf{h}}^s$, where we compute the diffusion loss $\mathcal{L}_{diff}$ of DD through

$$\mathcal{L}_{diff} = \mathbb{E}_{t \sim U(0,T)} \| f_{dd}(\mathbf{h}_t, \mathbf{h}^v, \widehat{\mathbf{h}}^s, t) - \mathbf{h}_0 \|_2^2 \tag{13}$$

Upon the reconstructed representation, we use a linear projection layer to predict the probability distribution over all tokens. Afterwards, we compute cross-entropy loss $\mathcal{L}_{CE}$ by comparing $\widehat{\mathcal{Y}}$ and $\mathcal{Y}^*$, where the final loss of DD $\mathcal{L}_{DD}$ is formulated by

$$\mathcal{L}_{DD} = \mathcal{L}_{CE} + \mathcal{L}_{diff} \tag{14}$$

**Diffusion Decoding** Diffusion decoding generates $\widehat{\mathcal{Y}}$ following the standard process of BD. Specifically, we randomly sample a Gaussian noise $\mathbf{n}$ and denoise it into the final representation $\widehat{\mathbf{h}}_0$ for $\widehat{\mathcal{Y}}$. In doing so, we initialize $\widehat{\mathbf{h}}_T$ with $\mathbf{n}$ and iteratively denoise it into $\widehat{\mathbf{h}}_0$ according to

$$\widehat{\mathbf{h}}_0 = \prod_{t=1}^{T} p(\widehat{\mathbf{h}}_{t-1} | \widehat{\mathbf{h}}_t, \mathbf{h}^v, \widehat{\mathbf{h}}^s) \tag{15}$$

where

$$p(\widehat{\mathbf{h}}_{t-1} | \widehat{\mathbf{h}}_t, \mathbf{h}^v, \widehat{\mathbf{h}}^s) = \sqrt{\bar{\alpha}_{t-1}} \cdot \frac{\widehat{\mathbf{h}}_t - \sqrt{1 - \bar{\alpha}_t} \cdot f_{dd}(\widehat{\mathbf{h}}_t, \mathbf{h}^v, \widehat{\mathbf{h}}^s, t)}{\sqrt{\bar{\alpha}_t}}$$
$$+ \sqrt{1 - \bar{\alpha}_{t-1}} \cdot f_{dd}(\widehat{\mathbf{h}}_t, \mathbf{h}^v, \widehat{\mathbf{h}}^s, t) \tag{16}$$

Finally, we decode $\widehat{\mathbf{h}}_0$ and obtain the final text results $\widehat{\mathcal{Y}}$ for the input image $\mathcal{I}$.

## 3 EXPERIMENT SETTINGS

### 3.1 DATASETS

We evaluate our approach on a series of datasets from different tasks, including MIMIC-CXR (Johnson et al., 2019) for RRG, CC-SBU (Zhu et al., 2023) for cross-modal alignment, Localized Narratives (LN) (Pont-Tuset et al., 2020) for IC. Details of the aforementioned datasets are reported in Table 1 and illustrated in the following text.

| DATASET | MIMIC-CXR | | | CC-SBU | | | LN | | | COCO-LT | | |
|---|---|---|---|---|---|---|---|---|---|---|---|---|
| | TRAIN | VAL | TEST | TRAIN | VAL | TEST | TRAIN | VAL | TEST | TRAIN | VAL | TEST |
| IMAGE | 369.0K | 3.0K | 5.2K | 3.0K | 0.1K | 0.3K | 1743K | 41.7K | 126.0K | 82.8K | 40.5K | 40.8K |
| DESCRIPTION | 222.8K | 1.8K | 3.3K | 3.0K | 0.1K | 0.3K | 507.4K | 41.7K | 126.0K | 48.8K | 3.0K | 3.0K |
| AVG. LEN. | 53.0 | 53.1 | 66.4 | 70.8 | 70.8 | 71.5 | 35.5 | 29.9 | 30.6 | 75.6 | 75.8 | 76.4 |

Table 1: Statistics of our experiment datasets w.r.t. their training, validation, and test sets, including the numbers of images, descriptions, and the average length of descriptions (i.e., (AVG. LEN.)).

| DATA | MODEL | NLG METRICS | | | | | | | CE METRICS | | |
|---|---|---|---|---|---|---|---|---|---|---|---|
| | | BL-1 | BL-2 | BL-3 | BL-4 | MTR | RG-L | AVG. $\Delta$ | P | R | F1 |
| MIMIC-CXR | TRANS | 0.357 | 0.216 | 0.141 | 0.091 | 0.129 | 0.271 | - | 0.348 | 0.314 | 0.330 |
| | DIFF | 0.380 | 0.221 | 0.143 | 0.100 | 0.137 | 0.277 | 4.5% | 0.385 | 0.401 | 0.393 |
| | +SCP | 0.409 | 0.243 | 0.167 | 0.113 | 0.149 | 0.284 | 12.8% | 0.437 | 0.445 | 0.441 |
| | +SCM | 0.385 | 0.227 | 0.149 | 0.106 | 0.142 | 0.279 | 6.3% | 0.405 | 0.417 | 0.411 |
| | +SCP+SCM (SEMDIFF) | **0.412**$^*$ | **0.259**$^*$ | **0.180**$^*$ | **0.129**$^*$ | **0.178**$^*$ | **0.287**$^*$ | **19.0%** | **0.471**$^*$ | **0.479**$^*$ | **0.478**$^*$ |
| CC-SBU | TRANS | 0.343 | 0.197 | 0.115 | 0.054 | 0.066 | 0.214 | - | - | - | - |
| | DIFF | 0.370 | 0.223 | 0.131 | 0.081 | 0.173 | 0.253 | 23.6% | - | - | - |
| | +SCP | 0.404 | 0.251 | 0.155 | 0.099 | 0.181 | 0.284 | 32.7% | - | - | - |
| | +SCM | 0.388 | 0.239 | 0.140 | 0.084 | 0.174 | 0.267 | 27.4% | - | - | - |
| | +SCP+SCM (SEMDIFF) | **0.417**$^*$ | **0.265**$^*$ | **0.167**$^*$ | **0.109**$^*$ | **0.201**$^*$ | **0.323**$^*$ | **37.7%** | - | - | - |
| LN | TRANS | 0.197 | 0.117 | 0.063 | 0.040 | 0.095 | 0.151 | - | - | - | - |
| | DIFF | 0.220 | 0.139 | 0.087 | 0.053 | 0.117 | 0.175 | 18.5% | - | - | - |
| | +SCP | 0.305 | 0.175 | 0.102 | 0.067 | 0.130 | 0.220 | 34.2% | - | - | - |
| | +SCM | 0.291 | 0.164 | 0.138 | 0.061 | 0.125 | 0.206 | 33.4% | - | - | - |
| | +SCP+SCM (SEMDIFF) | **0.376**$^*$ | **0.229**$^*$ | **0.148**$^*$ | **0.092**$^*$ | **0.153**$^*$ | **0.281**$^*$ | **49.1%** | - | - | - |
| COCO-LT | TRANS | 0.257 | 0.129 | 0.058 | 0.030 | 0.093 | 0.178 | - | - | - | - |
| | DIFF | 0.283 | 0.144 | 0.076 | 0.041 | 0.119 | 0.210 | 17.9% | - | - | - |
| | +SCP | 0.328 | 0.178 | 0.102 | 0.071 | 0.133 | 0.239 | 34.3% | - | - | - |
| | +SCM | 0.314 | 0.152 | 0.088 | 0.056 | 0.129 | 0.202 | 25.6% | - | - | - |
| | +SCP+SCM (SEMDIFF) | **0.365**$^*$ | **0.210**$^*$ | **0.144**$^*$ | **0.093**$^*$ | **0.155**$^*$ | **0.265**$^*$ | **44.7%** | - | - | - |

Table 2: Comparison of different baselines with the full model (SEMDIFF) on four datasets under NLG and CE metrics (CE only applies to MIMIC-CXR). "BL" denotes the abbreviation of BLEU; "MTR" and "RG-L" denote METEOR and ROUGE-L, respectively. The average improvement over all NLG metrics compared to "Trans" is also presented in the "AVG. $\Delta$" column. $^*$ marks the results where the improvements are statistically significant over all baselines at $p \leq 0.05$ level.

**MIMIC-CXR** is the largest public dataset for RRG with 473,057 chest X-Ray images and 206,563 reports. We follow its official split and utilize the medical text indexer (MTI)[2] to preprocess all radiology reports in obtaining medical concepts. **CC-SBU** is a dataset proposed by MiniGPT-4 (Zhu et al., 2023), which contains 3,439 high-quality image-description pairs. In this dataset, we use key words in image description as semantic concepts by filtering them according to their part-of-speech (POS) tags and frequencies. In doing so, we employ the NLTK POS tagger to annotate POS labels for each word in image descriptions[3] and set a threshold to filter out infrequent words. Based on the aforementioned process, we finally obtain 1,622 semantic concepts (words) for CC-SBU. For **Localized Narratives (LN)**, we choose its Open Images [4] subset containing 671k image-description pairs for our experiments and obtain the semantic concepts following the similar pipeline as that applied to CC-SBU, resulted in 4,888 semantic concepts (words) in total.

Particularly, we propose a new dataset **COCO-LT** dedicated to I2LTG based on COCO (Lin et al., 2014) for further evaluating our approach. In detail, we randomly choose around 40% of original COCO instances to form this dataset with each image in it having five corresponding short description sentences from different perspectives. Then we employ ChatGPT (GPT-3.5-Turbo) to produce a long description (generally a paragraph) based on these sentences through a special prompt and finally result in 54,785 image-description pairs. For this dataset, we utilize the similar process as that for CC-SBU and COCO-LT, and obtain 1,894 semantic concepts (words).[5]

---

[2]https://lhncbc.nlm.nih.gov/ii/tools/MTI.html

[3]Preserved POS labels only consist JJ, JJR, JJS, NN, NNS, RB, RBR, RBS, VB, VBD, VBG, VBZ.

[4]https://github.com/cvdfoundation/open-images-dataset

[5]We illustrate more details of the proposed COCO-LT dataset in Appendix A.

| Data | Model | NLG Metrics | | | | | | CE Metrics | | |
|---|---|---|---|---|---|---|---|---|---|---|
| | | BL-1 | BL-2 | BL-3 | BL-4 | MTR | RG-L | P | R | F1 |
| MIMIC-CXR | ST (Vinyals et al., 2015) | 0.299 | 0.184 | 0.121 | 0.084 | 0.124 | 0.263 | 0.249 | 0.203 | 0.204 |
| | ATT2IN (Rennie et al., 2017) | 0.325 | 0.203 | 0.136 | 0.096 | 0.134 | 0.276 | 0.322 | 0.239 | 0.249 |
| | ADAATT (Lu et al., 2017) | 0.299 | 0.185 | 0.124 | 0.088 | 0.118 | 0.266 | 0.268 | 0.186 | 0.181 |
| | TOPDOWN (Anderson et al., 2018) | 0.317 | 0.195 | 0.130 | 0.092 | 0.128 | 0.267 | 0.320 | 0.231 | 0.238 |
| | R2GEN (Chen et al., 2020) | 0.353 | 0.218 | 0.145 | 0.103 | 0.142 | 0.277 | 0.333 | 0.273 | 0.276 |
| | CA (Liu et al., 2021c) | 0.350 | 0.219 | 0.152 | 0.109 | 0.151 | 0.283 | - | - | - |
| | CMCL (Liu et al., 2021a) | 0.344 | 0.217 | 0.140 | 0.097 | 0.133 | 0.281 | - | - | - |
| | PPKED (Liu et al., 2021b) | 0.360 | 0.224 | 0.149 | 0.106 | 0.149 | 0.284 | - | - | - |
| | R2GENCMN (Chen et al., 2021) | 0.353 | 0.218 | 0.148 | 0.106 | 0.142 | 0.278 | 0.334 | 0.275 | 0.278 |
| | R2GENRL (Qin & Song, 2022) | 0.381 | 0.232 | 0.155 | 0.109 | 0.151 | 0.287 | 0.342 | 0.294 | 0.292 |
| | ITA (Wang et al., 2022) | 0.395 | 0.253 | 0.170 | 0.121 | 0.147 | 0.284 | - | - | - |
| | WARMSTART (Nicolson et al., 2023) | 0.392 | 0.245 | 0.169 | 0.124 | 0.153 | 0.285 | 0.359 | 0.412 | 0.384 |
| | RGRG (Tanida et al., 2023) | 0.373 | 0.249 | 0.175 | 0.126 | 0.168 | 0.264 | 0.461 | 0.475 | 0.447 |
| | ORGAN (Hou et al., 2023) | 0.407 | 0.256 | 0.172 | 0.123 | 0.162 | 0.293 | 0.416 | 0.418 | 0.385 |
| | KIUT (Huang et al., 2023) | 0.393 | 0.243 | 0.159 | 0.113 | 0.160 | 0.285 | 0.371 | 0.318 | 0.321 |
| | XRAYGPT (7B) (Omkar Thawkar & Khan, 2023) | 0.128 | 0.045 | 0.014 | 0.004 | 0.079 | 0.111 | - | - | - |
| | MED-PALM (562B) (Tu et al., 2023) | 0.317 | - | - | 0.115 | - | 0.275 | - | - | 0.378 |
| | SEMDIFF | 0.412* | 0.259* | 0.180* | 0.129* | 0.178* | 0.287* | 0.471* | 0.479* | 0.478* |

Table 3: Comparisons of SEMDIFF with previous studies on the test set of MIMIC-CXR under NLG and CE metrics. The best and second results are in boldface and underlined. For LLM-based methods (i.e., XRAYGPT, MED-PALM), we also illustrate their parameter numbers in parentheses. * marks the results the improvements are statistically significant over all baselines at $p \leq 0.05$ level.

## 3.2 BASELINES AND EVALUATION METRICS

To verify our proposed model, we use four baselines for comparison in our experiments. "Trans" represents the autoregressive model with ResNet-101 (He et al., 2016) and a 3-layer Transformer as the visual encoder, and another 3-layer Transformer with an additional 8-head cross-attention layer as the decoder, and "Diff" denotes our baseline diffusion model which directly generates the image description from the visual representations. "+SCP" stand for the model that SCP is applied to "Diff", serving as our third baseline. "+SCM" represents our fourth baseline model that "Diff" is equipped with only SCM, where SCM directly interacts with visual representations. "+SCP+SCM" is our full model with all proposed components.

For evaluation on MIMIC-CXR, we follow previous studies (Chen et al., 2020; 2021; Qin & Song, 2022; Huang et al., 2023) and evaluate the different models with natural language generation (NLG) and clinical efficacy (CE) metrics. For NLG metrics, we use BLEU (Papineni et al., 2002), ME-TEOR (Michael & Alon, 2011), and ROUGE-L (Lin, 2004). For CE metrics, we employ CheXpert (Gao et al., 2019b) to classify words in the generated reports into 14 different categories related to thoracic diseases and support devices, and compare the resulted labels with the ones in gold standard reports. We use precision, recall, and F1 to evaluate model performance for CE metrics. For evaluation on CC-SBU, LN, COCO-LT, we only use NLG metrics following conventional studies (Vinyals et al., 2015; Rennie et al., 2017; Anderson et al., 2018; Cornia et al., 2020; Fang et al., 2022; Li et al., 2022b) and also measure the lengths of the generated texts.

## 3.3 IMPLEMENTATION DETAILS

In our experiments, we try different hyper-parameter settings and select the one with best perfor-mance on the validation set. [6] For model architecture, we implement $f_{ve}$, $f_{scp}$, and $f_{dd}$ with 3 layers of Transformer, where number of the attention head and dimension of the hidden vectors are set to 8 and 512, respectively. In SCP and DD, we implement an additional 8-head cross-attention layer to incorporate the visual representations. For SCM, the memory dimension $d$ is set to 512. For DD, the total step $T$ for diffusion forwarding and decoding processes is set to 100. For optimization, we use Adam (Kingma & Ba, 2015) optimizer updating all model parameters with a learning rate of 5e-4. We follow the learning rate scheduling strategy in Vaswani et al. (2017) with 20,000 steps for warm-up, where the total training steps vary from 1.5M to 6.7M according to different datasets. The weights to balance SCP and DD loss in Eq. 2 are set to $\beta_1 = 1$ and $\beta_2 = 1$, respectively.

## 4 RESULTS AND ANALYSIS

### 4.1 OVERALL RESULTS

Experimental results of different models on the test sets of four datasets are reported in Table 2, with several observations. First, in all four test sets, it is observed that the basic non-AR model

---

[6]We report the details of hyper-parameter settings in Appendix B.

| MODEL | CC-SBU | | | LN | | | COCO-LT | | |
|---|---|---|---|---|---|---|---|---|---|
| | BL-4 | MTR | RG-L | BL-4 | MTR | RG-L | BL-4 | MTR | RG-L |
| ATT2IN (Rennie et al., 2017) | 0.0001 | 0.0114 | 0.0573 | 0.0002 | 0.0138 | 0.0628 | 0.0001 | 0.0079 | 0.0518 |
| RFNET (Jiang et al., 2018) | 0.0001 | 0.0126 | 0.0557 | 0.0002 | 0.0144 | 0.0731 | 0.0001 | 0.0066 | 0.0489 |
| TOPDOWN (Anderson et al., 2018) | 0.0002 | 0.0238 | 0.0694 | 0.0002 | 0.0177 | 0.0749 | 0.0001 | 0.0224 | 0.0545 |
| MIR (Lee et al., 2018) | 0.0002 | 0.0238 | 0.0694 | 0.0002 | 0.0162 | 0.0713 | 0.0002 | 0.0365 | 0.0894 |
| ORT (Herdade et al., 2019) | 0.0003 | 0.0519 | 0.1031 | 0.0003 | 0.0257 | 0.0861 | 0.0002 | 0.0438 | 0.1268 |
| AOANET (Huang et al., 2019) | 0.0002 | 0.0486 | 0.0933 | 0.0003 | 0.0286 | 0.0875 | 0.0002 | 0.0457 | 0.1381 |
| $M^2$ TRANSFORMER (Cornia et al., 2020) | 0.0002 | 0.0466 | 0.0859 | 0.0004 | 0.0347 | 0.0914 | 0.0002 | 0.0432 | 0.1219 |
| SATIC (Zhou et al., 2021) | 0.0003 | 0.0530 | 0.1182 | 0.0007 | 0.0704 | 0.1462 | 0.0003 | 0.0469 | 0.1227 |
| SCD-NET (Luo et al., 2022) | 0.0003 | 0.0451 | 0.1238 | 0.0006 | 0.0549 | 0.1201 | 0.0001 | 0.0429 | 0.1374 |
| BLIP-2 (1.1B) (Li et al., 2023) | 0.0017 | 0.0557 | 0.1552 | 0.0013 | 0.0430 | 0.1050 | 0.0004 | 0.0476 | 0.1257 |
| MINIGPT-4 (13B) (Zhu et al., 2023) | 0.0875 | 0.1561 | 0.2256 | 0.0153 | 0.1120 | 0.1478 | 0.0211 | 0.1008 | 0.1441 |
| LLAVA (7B) (Liu et al., 2023) | 0.0603 | 0.1626 | 0.2467 | 0.0155 | 0.1333 | 0.1856 | 0.0386 | 0.1441 | 0.2010 |
| SEMDIFF | **0.1088**$^*$ | **0.2007**$^*$ | **0.3229**$^*$ | **0.0920**$^*$ | **0.1533**$^*$ | **0.2814**$^*$ | **0.0934**$^*$ | **0.1547**$^*$ | **0.2649**$^*$ |

Table 4: Comparisons of our approach with previous studies on the test sets of CC-SBU, LN, and COCO-LT under NLG metrics (BL, MTR and RG refer to BLEU, METEOR and ROUGE, respectively). The best and second results are in boldface and underlined. LLM-based methods (i.e., BLIP-2, MINIGPT-4, and LLAVA) are illustrated with their parameter numbers in parentheses. $^*$ marks the results where improvements are statistically significant at $p \leq 0.05$ level over all baselines.

("Diff") consistently outperforms the AR one ("Trans") on all datasets, owing to that the error propagation problem is alleviated. Second, by comparing whether using semantic information, "Diff+SCP" (i.e., latent representations and explicit semantic concepts) leads to significantly better performance over "Diff" (i.e., latent representations), which confirms the effectiveness of semantic guidance for I2LTG.[7] Third, comparing approaches with and without using memory, we find that "Diff+SCM" achieves better performance than "Diff", which indicates that SCM helps the model to establish a better cross-modal alignment. Fourth, when SCP and SCM are combined, our approach "Diff+SCP+SCM" is able to further enhance the performance of "Diff+SCP" and "Diff+SCM", and achieves the best result, which indicates the necessity to optimize semantic concepts in SCM.

To further illustrate the effectiveness of our approach, we compare it with existing state-of-the-art solutions[8] on all four datasets, with results presented in Table 3 and 4.[9] Overall, our approach significantly outperforms other approaches on all metrics, which illustrates the superiority of our approach for I2LTG with its specific model design. Notably, our approach even achieves better performance than those studies based on large language models (LLMs) (i.e., XRAYGPT, MED-PALM[10], BLIP-2, MINIGPT-4, and LLAVA), indicating that appropriate semantic guidance is more efficient than using a massive amount of parameters in LLMs. Compared to prevailing non-AR solutions (i.e., MIR, SATIC, and SCD-NET), our approach obtains significant improvements, suggesting the power of semantic concepts in helping non-AR models with overcoming their limitations such as word repetition issue, which are further illustrated in the next subsection. Particularly, in noticing that SCD-NET also leverages semantic guidance, our approach presents its superior capability in generating better results by utilizing predicted semantic concepts while SCD-NET obtains such semantic information by retrieving and encoding sentences, resulting in a coarser guidance.

## 4.2 ANALYSIS

We perform a series of analysis to investigate the effect of different components of our approach following its pipeline sequence. Specifically, we firstly explore how semantic matrix size affects the concept prediction process in SCP. Then, we investigate SCM performance against different memory sizes and the number of queried memory vectors. Finally, we qualitatively illustrate the effect of different components of our approach through a case study.

**Effect of the Semantic Matrix Size** We conduct our approach with different semantic matrix sizes (i.e., $\mathcal{N}_s$) to analyze their effects to SCP. Figure 2 (a) presents the curves of BLEU-4 score against

---

[7]To comprehensively evaluate the quality of the semantic guidance, we compare the generated concepts with the ones in gold standard descriptions, and present the results (precision, recall, and F1) in Appendix C.

[8]The guideline for choosing these studies is based on that they have open-sourced code, which allows us to run their models on our experiment datasets, especially the COCO-LT dataset proposed in this paper.

[9]We report full evaluation with all metrics on our approach and existing state-of-the-art solutions on CC-SBU, LN, and COCO-LT datasets in Appendix D.

[10]MED-PALM does not release the model weights and its RRG test set. Therefore, for fair comparisons, we approximate their settings to randomly curated 10 groups of test instances with the same size (i.e., 246 cases) as that used in MED-PALM. Under this setting, SEMDIFF performs similarly to the results reported in Table 3.

Figure 2: The curves of BLEU-4 score on test sets of different datasets with respect to (a) semantic matrix size, (b) memory size, and (c) number of queried memory vectors.

$\mathcal{N}_s$, showing that the semantic matrix size should be seperately set for different datasets. In general, when this size is smaller than the optimal value, the model gradually obtains better performance as $\mathcal{N}_s$ increases, which indicates that semantic matrix is able to cover more related concepts so that SCP stores more essential semantic information. However, once the optimal value is reached, model performance starts to degrade when the size keeps enlarging, thus overfitting is observed accordingly and larger matrix size does not help in storing useful semantic information.

**Effect of the Memory Size**    To explore the effect of memory size on SCM (i.e., $\mathcal{N}_m$), we conduct our approach with different $\mathcal{N}_m$. Figure 2 (b) presents the curves of BLEU-4 score with respect to $\mathcal{N}_m$ ranging from 32 to 4,096. It is observed that, in general, enlarging the memory matrix helps improving model performance on all datasets, indicating that better generation results are expected when a larger matrix is applied and stores more image-text correlation information. Moreover, we also notice performance convergence when $\mathcal{N}_m$ reaches 2048 (512 on CC-SBU), so that there exists a limit for the bonus on enlarging matrix size for preserving essential information.

**Effect of the Number of Queried Memory Vectors**    In analyzing how the number of queried memory vectors (i.e., $\mathcal{K}$) affects the SCM, we try our approach under different $\mathcal{K}$ settings. Figure 2 (c) presents the curves of BLEU-4 score with respect to $\mathcal{K}$ ranging from 1 to 512. Similar to that found in semantic matrix size analysis, it is shown that $\mathcal{K}$ has an optimal value on each dataset, where retrieving either too few or too many memory vectors leads to inferior performance, corresponding to the situations of information insufficiency and overloading, respectively. Particularly, when too many vectors are retrieved, the impact of noise is highly significant in affecting model performance as the BLEU-4 scores rapidly drop, suggesting that $\mathcal{K}$ should be carefully chosen.

**Case Study**    In addition to quantitative analyses, we also present a case study on the generated texts from different models with the same image input from CC-SBU. Figure 3 demonstrates the results with comparison of iterative generations from "Diff" and "Diff+SCP+SCM", where semantic words shared by model outputs and the gold standard texts are highlighted in the same color, as well as the time step $t$ in iteration and the average number of repetitive words in different results illustrated in parentheses.[11] There are several observations from different perspectives. "Diff" gradually refines the initialized repetitive words into a series of descriptive sentences, which produces few related semantic words in its results, suggesting the ambiguity of visual representation that leads to insufficient semantic information for the text generation process. On the contrary, with the assistance of semantic concepts, our full model ("Diff+SCP+SCM") is able to generate more reasonable results that contain enough related contents, indicating that SCP and SCM provide a strong guidance for the generation process to produce semantic coherent long texts. Notably, "Diff+SCP+SCM" also performs a more organized generation process, where the number of repetitive words is significantly decreased during the iterative generation process, which confirms the validity of our model design and the potential of semantic concepts to alleviate existing limitations of non-AR solutions.[12]

## 5 RELATED WORK

Conventionally, describing images is primarily carried out through image captioning (IC), where normally short sentences are generated for input source images based on autoregressive models (i.e., LSTM (Hochreiter & Schmidhuber, 1997; Vaswani et al., 2017)) or non-autoregressive ones (Lee et al., 2018; Gao et al., 2019a; Guo et al., 2020; Zhou et al., 2021), with pre-training techniques (Hu et al., 2022; Chen et al., 2022; Nukrai et al., 2022; Romain & Rufin, 2023; Ramos et al., 2023), semantic condition (Fang et al., 2022; Li et al., 2022b), and enhanced multi-modal features (Shi et al., 2021; Ng et al., 2021; Nguyen et al., 2022; Liu et al., 2022; Wu et al., 2022; Zhang et al., 2022; Wu et al., 2023) applied to facilitate the generation process. However, IC normally fails

---

[11]We further report word repetition results from different models on all datasets in Appendix E.

[12]For comprehensive comparisons, we present more case studies in Appendix F.

| Image | Diff | Diff+SCP+SCM |
|---|---|---|
| *Gold Standard*

The image shows a view from the **top** of **mountains**, looking down into the **valley** below. The **valley** is lined with **trees** and there are **mountains** in the distance. The **sky** is **clear** with some **clouds** in the distance, and the **sun** is **shining** down on the scene. | **(t=100, 52.0)** *The image image objective objective ... objective* | **(t=100, 14.3)** *The image shows shows **top** top mountains ... mountains objective ... objective valley valley green green green green **trees** trees ..trees **sky** sky sky **clear** clear ... clear blue blue objective **clouds** ... clouds **sun** ... sun **shining** shining shining* |
| | **(t=75, 19.0)** *The image shows a view view of objective ... objective **mountains** in a of the ... the in in behind ... behind* | **(t=75, 6.9)** *The image shows a view on **top** top **mountains** mountains mountains with **valley** valley and green green **trees** trees. The **sky** sky is **clear** clear and blue with objective and **clouds** clouds of ... of sun is **shining** shining shining shining 500 scene* |
| | **(t=50, 6.4)** *The image shows a view of a 6 well ... of a mountains with leather 52 ... there is a white objective sky with objectivery ... the water in the of a mountains is crew and frame* | **(t=50, 3.1)** *The image shows a view on **top** of **mountains** of a3 the ... with **valley** and green green **trees** in the of a ... The **sky** is **clear** and blue b60\*lue with **clouds** objective 7 website ...and clouds clouds. The **sun** is **shining** shining on objective of a scene.* |
| | **(t=25, 1.6)** *The image shows a view of rocky **valley** **mountains** mountains There is a white blue **sky** mountains with **clouds** mountains and the water moutains behind the mountains is **clear** and blue and mountains in the distance* | **(t=25, 1.4)** *The image shows a view on **top** of rocky **mountains,** with a **valley** hill and the green **trees** on a large amount of **mountains** in the distance. The **sky** is **clear** and blue with **clouds**. The **sun** is **shining** on the side of the **valley** scene.* |
| | **(t=0, 1.5)** *The image shows a small view of a rocky **valley** and **mountains**. There is a blue **sky** with **clouds** on top of the moutains. The water behind the mountains is **clear** and blue. The mountains and valley are in the distance.* | **(t=0, 1.3)** *The image shows a view on **top** of rocky **mountains,** with the **valley** surrounded by green **trees.** There is large amount of **mountains** in the distance. The **sky** is **clear** and blue with **clouds**. The **sun** is **shining** on the **valley** scene.* |

Figure 3: An illustration of the text generation processes (through texts generated at different steps) by different models with an example input image from CC-SBU, where the gold standard is also presented for reference. Semantic words (i.e., semantic concepts) shared by model outputs and the gold standard texts are highlighted in the same color. $t$ refers to the step of the diffusion decoding process, where $t$ decreases from 100 to 0 (following the convention setting of diffusion models, step decreasing represents the iterative process starting from Gaussian noises to final de-noised results).

to meet the requirements of some challenging scenarios, especially the ones in particular domain with long descriptions, e.g., report for radiology. Although some approaches directly use IC models (Vinyals et al., 2015; Lu et al., 2017; Rennie et al., 2017; Anderson et al., 2018) for radiology report generation (RRG), some studies improve conventional AR solutions with co-attentions (Jing et al., 2018), memory networks (Chen et al., 2020; 2021), reinforcement learning (Qin & Song, 2022), and useful features in different modalities (Li et al., 2018; Wang et al., 2022; Tanida et al., 2023; Hou et al., 2023; Huang et al., 2023), which are still limited to guarantee comprehensive and coherent texts in the generated result. With recent advances in large language models (LLMs) (Touvron et al., 2023a;b) and diffusion model (Ho et al., 2020) that both illustrate outstanding generation ability, these techniques have been employed to enhance the cross-modal content generation process (Li et al., 2023; Zhu et al., 2023; Liu et al., 2023) as well as report generation in the medical domain (Omkar Thawkar & Khan, 2023; Tu et al., 2023). Particularly, owing to the discrete nature of texts, it is hard to directly applying standard diffusion model for text generation, some studies are thus proposed to do so through continuous representations, e.g., embedding (Li et al., 2022a; Gong et al., 2023) and bit representations (Chen et al., 2023; Luo et al., 2022). Compared with all aforementioned work, our approach offers a generic solution for I2LTG, with an effective design of using diffusion networks for non-AR text generation, and proves the validity of employing semantic guidance to enhance the coherence of texts when generating long descriptions for an image.

## 6 CONCLUSION

In this paper, we propose a diffusion-based model, SEMDIFF, with memory networks for I2LTG, which firstly captures salient semantic concepts in image, then utilizes memory networks to enhance such concepts, and finally employs diffusion networks to incorporate them to facilitate the long-text generation process. SEMDIFF offers a solution to incorporating external guidance into diffusion networks, effectively addresses a series of issues such as incoherence problem in non-AR text generation, especially for long texts. Experiments on three public datasets and COCO-LT illustrate the superiority of our approach compared to state-of-the-art solutions. We also propose a new dataset COCO-LT dataset with over 54K image-long text pairs to further evaluate our approach on I2LTG, which further confirms its long-text generation ability as that proved on the three public datasets. Further analyses investigate the effect of our approach in accommodating semantic concepts into diffusion networks, indicating that our SEMDIFF design of incorporating external guidance has its potential of being utilized as a benchmark framework for similar tasks in future studies.

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

> **Prompt:**
> *<sentence-1> <split> <sentence-2> <split> <sentence-3> <split> <sentence-4> <split> <sentence-5>*
> *Conclude the above sentences into one paragraph.*

Figure 4: The prompt we used for ChatGPT to generate long text description. "$<sentence\text{-}i>$" represents the placeholder of the $i$-th ($i \in \{1 \ldots 5\}$) COCO caption given the same image and "$<split>$" denotes the special token to mark the boundaries between every two sentences.

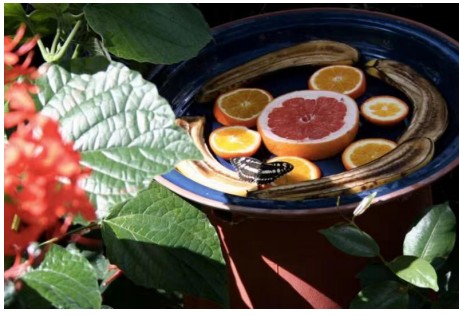

**(a) Image**

> *1. A bowl in the garden filled with fruit for the animals.*
> *2. Sliced grape fruit and oranges are placed in a bowl with a butterfly on one of the oranges.*
> *3. A plate with sliced fruit has attached a butterfly.*
> *4. A plate fool of sliced oranges on a table, next to plants.*
> *5. A butterfly exploring a dish of fruits in the shade.*

**(b) COCO Captions**

> *In the garden, there is a bowl filled with sliced grapefruit and oranges, specifically placed to provide fruit for the animals. One of the oranges in the bowl has a butterfly resting on it adding a touch of beauty to the arrangement. Additionally, on a nearby table, there is a plate filled with sliced oranges, accompanied by plants. A butterfly can be seen exploring the dish of fruits, adding a sense of wonder to the scene.*

**(c) COCO-LT Description**

Figure 5: An illustration of (a) the image, (b) the original captions in COCO, and (c) the description generated by ChatGPT from COCO captions for COCO-LT.

## APPENDIX A: MORE DETAILS OF THE COCO-LT DATASET

When creating the COCO-LT dataset, we prompt ChatGPT to generate long text descriptions using the five captions from the instances in original COCO dataset. We show the prompt used by ChatGPT in Figure 4 and present an example in the COCO-LT dataset with the image, the original COCO captions, and the produced descriptions in Figure 5 (a), (b), and (c), respectively.

## APPENDIX B: HYPER-PARAMETER SETTINGS

We report the detailed hyper-parameter settings for different datasets in Table 5. For the bit dimension, we follow the standard process of Bit Diffusion (BD) (Chen et al., 2023) and set the value according to the vocabulary size of each dataset. Herein, we choose a frequency threshold according to the vocabulary size of each dataset, where the dataset with a smaller vocabulary has a higher threshold. For example, since MIMIC-CXR has the smallest vocabulary, we set its threshold to $80$ in order to obtain accurate medical concepts in its radiology reports. We also report numbers of

| | MIMIC-CXR | CC-SBU | LN | COCO-LT |
|---|---|---|---|---|
| Bit Dimension | 13 | 14 | 14 | 14 |
| Frequency Threshold | 80 | 10 | 30 | 30 |
| Memory Size $N_m$ | 2,048 | 512 | 2,048 | 2,048 |
| Number of Queried Vectors $\mathcal{K}$ | 128 | 64 | 128 | 128 |
| Training Epochs | 30 | 2,000 | 10 | 30 |
| Concept Size | 931 | 1,622 | 4,888 | 1,894 |

Table 5: Details of hyper-parameter settings for MIMIC-CXR, CC-SBU, LN, and COCO-LT. The number of concepts obtained by applying the frequency threshold to each dataset is also reported.

| DATA | P | R | F1 |
|---|---|---|---|
| MIMIC-CXR | 0.764 | 0.649 | 0.695 |
| CC-SBU | 0.528 | 0.571 | 0.549 |
| LN | 0.469 | 0.513 | 0.490 |
| COCO-LT | 0.395 | 0.459 | 0.425 |

Table 6: Evaluation of the semantic concepts generated by SCP on the test sets of MIMIC-CXR, CC-SBU, LN, and COCO-LT with respect to precision (P), recall (R), and F1 scores.

concept based on the frequency threshold setting in Table 5. For the memory size $\mathcal{N}_m$, the number of queried vectors $\mathcal{K}$, and the training epochs of each dataset, we try different settings and select their combination with the best performance on the validation set. We choose $\mathcal{N}_m$ and $\mathcal{K}$ according to the analyses in Section 4.2, and set the training epochs based on the scale of each dataset.

## APPENDIX C: EVALUATION OF THE QUALITY OF PREDICTED SEMANTIC CONCEPTS

To evaluate the quality of the semantic guidance in our approach, we compare the semantic concepts generated by the SCP with gold standards, where the precision, recall, and F1 scores are reported in Table 6. The high F1 scores indicate a promising quality of the generated concepts.

## APPENDIX D: COMPREHENSIVE RESULTS ON CC-SBU, LN, AND COCO-LT

In Table 4, we only report BLEU-4, METEOR, ROUGE-L of different models on CC-SBU, LN, and COCO-LT datasets. For the performance on other metrics (i.e., BLEU-1, BLEU-2, and BLEU-3), we report them in Table 7, also with results on BLEU-4, METEOR, ROUGE-L presented for a comprehensive evaluation. From Table 7, we observe a similar trend as the one in Table 4, which further confirms the effectiveness of our approach. In addition, we present the average token-based length (i.e., "LEN") of the descriptions generated by different approaches to show their long text generation ability. We find that our approach is able to generate longer descriptions than conventional AR (e.g., ORT, M$^2$ TRANSFORMER) and non-AR approaches (e.g., "MIR", "SATIC", and "SCD-NET") with more coherent content (as that compared in NLG metrics). Moreover, compared with LLM-based methods (i.e., BLIP-2, MINIGPT-4, and LLAVA), although they are able to produce longer descriptions, our approach consistently outperforms them under all evaluation metrics, indicating a stronger generation ability with our model design than using massive parameters.

## APPENDIX E: WORD REPETITION RESULTS

We count the number of repeated tokens in the texts generated by different models and report their average numbers for each dataset in Table 8. We observe that, models with SCP (i.e., "Diff+SCP" and "Diff+SCP+SCM") are able to generate texts with less repetition than "SCD-NET" and "DIFF", which confirms the effectiveness of SCP in addressing the conventional non-AR generation issue.

| Data | Model | NLG Metrics | | | | | | |
|---|---|---|---|---|---|---|---|---|
| | | BL-1 | BL-2 | BL-3 | BL-4 | MTR | RG-L | Len |
| CC-SBU | ATT2IN (Rennie et al., 2017) | 0.0005 | 0.0003 | 0.0001 | 0.0001 | 0.0114 | 0.0573 | 6.9 |
| | RFNET (Jiang et al., 2018) | 0.0006 | 0.0002 | 0.0001 | 0.0001 | 0.0126 | 0.0557 | 7.2 |
| | TOPDOWN (Anderson et al., 2018) | 0.0010 | 0.0006 | 0.0003 | 0.0002 | 0.0238 | 0.0694 | 8.5 |
| | MIR (Lee et al., 2018) | 0.0014 | 0.0008 | 0.0004 | 0.0002 | 0.0283 | 0.0771 | 10.9 |
| | ORT (Herdade et al., 2019) | 0.0019 | 0.0011 | 0.0006 | 0.0003 | 0.0519 | 0.1031 | 11.5 |
| | AOANET (Huang et al., 2019) | 0.0022 | 0.0010 | 0.0006 | 0.0002 | 0.0486 | 0.0933 | 11.4 |
| | M$^2$ TRANSFORMER (Cornia et al., 2020) | 0.0016 | 0.0008 | 0.0005 | 0.0002 | 0.0466 | 0.0859 | 10.4 |
| | SATIC (Zhou et al., 2021) | 0.0018 | 0.0009 | 0.0006 | 0.0003 | 0.0530 | 0.1182 | 11.8 |
| | SCD-NET (Luo et al., 2022) | 0.0017 | 0.0007 | 0.0005 | 0.0002 | 0.0451 | 0.1238 | 13.4 |
| | BLIP-2 (1.1B) (Li et al., 2023) | 0.0044 | 0.0031 | 0.0022 | 0.0017 | 0.0557 | 0.1552 | 11.2 |
| | MINIGPT-4 (13B) (Zhu et al., 2023) | 0.3379 | 0.2024 | 0.1293 | 0.0875 | 0.1561 | 0.2256 | 60.5 |
| | LLAVA (7B) (Liu et al., 2023) | 0.3428 | 0.1857 | 0.1019 | 0.0603 | 0.1626 | 0.2467 | **72.8** |
| | SEMDIFF | **0.4172** | **0.2649** | **0.1674** | **0.1088** | **0.2007** | **0.3229** | 53.7 |
| LN | ATT2IN (Rennie et al., 2017) | 0.0022 | 0.0012 | 0.0003 | 0.0002 | 0.0138 | 0.0628 | 7.1 |
| | RFNET (Jiang et al., 2018) | 0.0025 | 0.0013 | 0.0004 | 0.0002 | 0.0144 | 0.0731 | 8.2 |
| | TOPDOWN (Anderson et al., 2018) | 0.0031 | 0.0012 | 0.0003 | 0.0002 | 0.0177 | 0.0749 | 6.6 |
| | MIR (Lee et al., 2018) | 0.0034 | 0.0011 | 0.0005 | 0.0002 | 0.0162 | 0.0713 | 10.2 |
| | ORT (Herdade et al., 2019) | 0.0039 | 0.0015 | 0.0005 | 0.0003 | 0.0257 | 0.0861 | 10.9 |
| | AOANET (Huang et al., 2019) | 0.0038 | 0.0016 | 0.0006 | 0.0003 | 0.0286 | 0.0875 | 10.3 |
| | M$^2$ TRANSFORMER (Cornia et al., 2020) | 0.0042 | 0.0018 | 0.0008 | 0.0004 | 0.0347 | 0.0914 | 11.4 |
| | SATIC (Zhou et al., 2021) | 0.0067 | 0.0038 | 0.0019 | 0.0007 | 0.0704 | 0.1462 | 12.1 |
| | SCD-NET (Luo et al., 2022) | 0.0057 | 0.0023 | 0.0012 | 0.0006 | 0.0549 | 0.1201 | 12.6 |
| | BLIP-2 (1.1B) (Li et al., 2023) | 0.0131 | 0.0058 | 0.0027 | 0.0013 | 0.0430 | 0.1050 | 10.6 |
| | MINIGPT-4 (13B) (Zhu et al., 2023) | 0.2045 | 0.0885 | 0.0359 | 0.0153 | 0.1120 | 0.1478 | 63.8 |
| | LLAVA (7B) (Liu et al., 2023) | 0.2083 | 0.0866 | 0.0348 | 0.0155 | 0.1333 | 0.1856 | **75.4** |
| | SEMDIFF | **0.3758** | **0.2290** | **0.1477** | **0.0920** | **0.1533** | **0.2814** | 55.8 |
| COCO-LT | ATT2IN (Rennie et al., 2017) | 0.0004 | 0.0002 | 0.0001 | 0.0001 | 0.0079 | 0.0518 | 6.5 |
| | RFNET (Jiang et al., 2018) | 0.0003 | 0.0002 | 0.0001 | 0.0001 | 0.0066 | 0.0489 | 6.4 |
| | TOPDOWN (Anderson et al., 2018) | 0.0008 | 0.0003 | 0.0002 | 0.0001 | 0.0224 | 0.0545 | 8.2 |
| | MIR (Lee et al., 2018) | 0.0010 | 0.0004 | 0.0002 | 0.0002 | 0.0365 | 0.0894 | 11.2 |
| | ORT (Herdade et al., 2019) | 0.0013 | 0.0006 | 0.0004 | 0.0002 | 0.0438 | 0.1268 | 10.5 |
| | AOANET (Huang et al., 2019) | 0.0015 | 0.0008 | 0.0004 | 0.0002 | 0.0457 | 0.1381 | 10.4 |
| | M$^2$ TRANSFORMER (Cornia et al., 2020) | 0.0013 | 0.0007 | 0.0003 | 0.0002 | 0.0432 | 0.1219 | 9.9 |
| | SATIC (Zhou et al., 2021) | 0.0017 | 0.0010 | 0.0006 | 0.0003 | 0.0469 | 0.1227 | 11.6 |
| | SCD-NET (Luo et al., 2022) | 0.0009 | 0.0007 | 0.0003 | 0.0001 | 0.0429 | 0.1374 | 13.7 |
| | BLIP-2 (1.1B) (Li et al., 2023) | 0.0016 | 0.0010 | 0.0006 | 0.0004 | 0.0476 | 0.1257 | 10.7 |
| | MINIGPT-4 (13B) (Zhu et al., 2023) | 0.2160 | 0.0968 | 0.0437 | 0.0211 | 0.1008 | 0.1441 | 60.4 |
| | LLAVA (7B) (Liu et al., 2023) | 0.3287 | 0.1579 | 0.0760 | 0.0386 | 0.1441 | 0.2010 | **73.9** |
| | SEMDIFF | **0.3654** | **0.2103** | **0.1435** | **0.0934** | **0.1547** | **0.2649** | 54.9 |

Table 7: Comparisons of our approach and existing state-of-the-art studies on CC-SBU, LN, and COCO-LT datasets w.r.t. all NLG metrics and averaged length (i.e., "LEN") of generated texts.

| Model | MIMIC-CXR | CC-SBU | LN | COCO-LT |
|---|---|---|---|---|
| SCD-NET (Luo et al., 2022) | - | 2.4 | 2.5 | 2.8 |
| DIFF | 3.5 | 3.8 | 3.1 | 4.9 |
| DIFF+SCP | 1.5 | 1.5 | 1.8 | 1.7 |
| DIFF+SCP+SCM (SEMDIFF) | 1.3 | 1.2 | 1.5 | 1.3 |

Table 8: The average number of repeated tokens in the descriptions produced by different models on MIMIC-CXR, CC-SBU, LN, and COCO-LT datasets.

## APPENDIX F: MORE CASE STUDIES

To further illustrate the effectiveness of our approach with qualitative comparison, we present more case studies on MIMIC-CXR, LN, and COCO-LT in Figure 6, 7, and 8, respectively, where the texts generated by all baselines and our full model, i.e., "Trans", "Diff", "Diff+SCP", "Diff+SCM", and "Diff+SCP+SCM". Similar to the observations in Figure 3, our full model is able to effectively leverage the semantic concepts generated and enhanced by SCP and SCM, respectively, so as to produce more comprehensive and coherent descriptions compared with other baseline models.

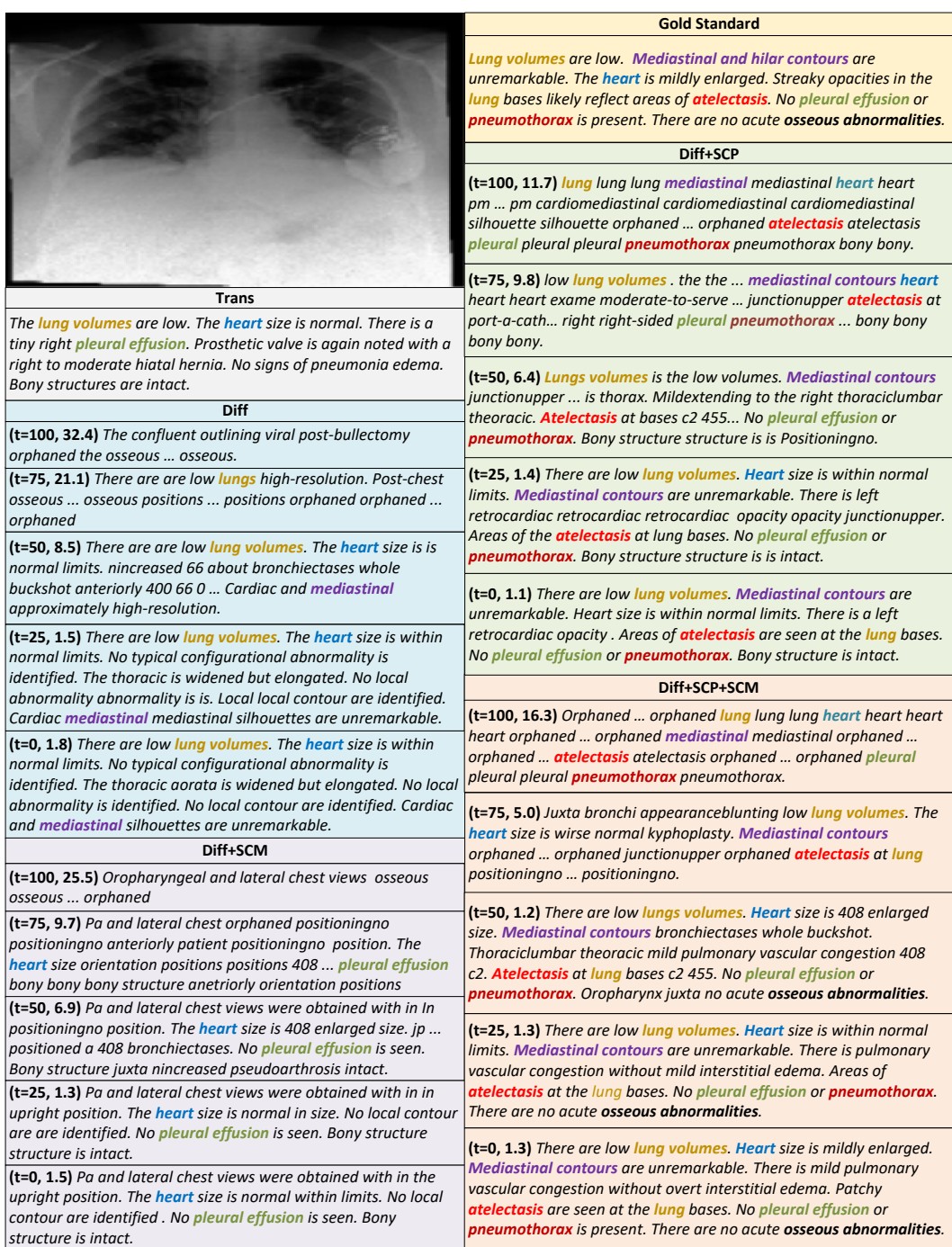

Figure 6: Descriptions generated by baseline models and our full model with an example input image from MIMIC-CXR, where the gold standard is also presented for reference.

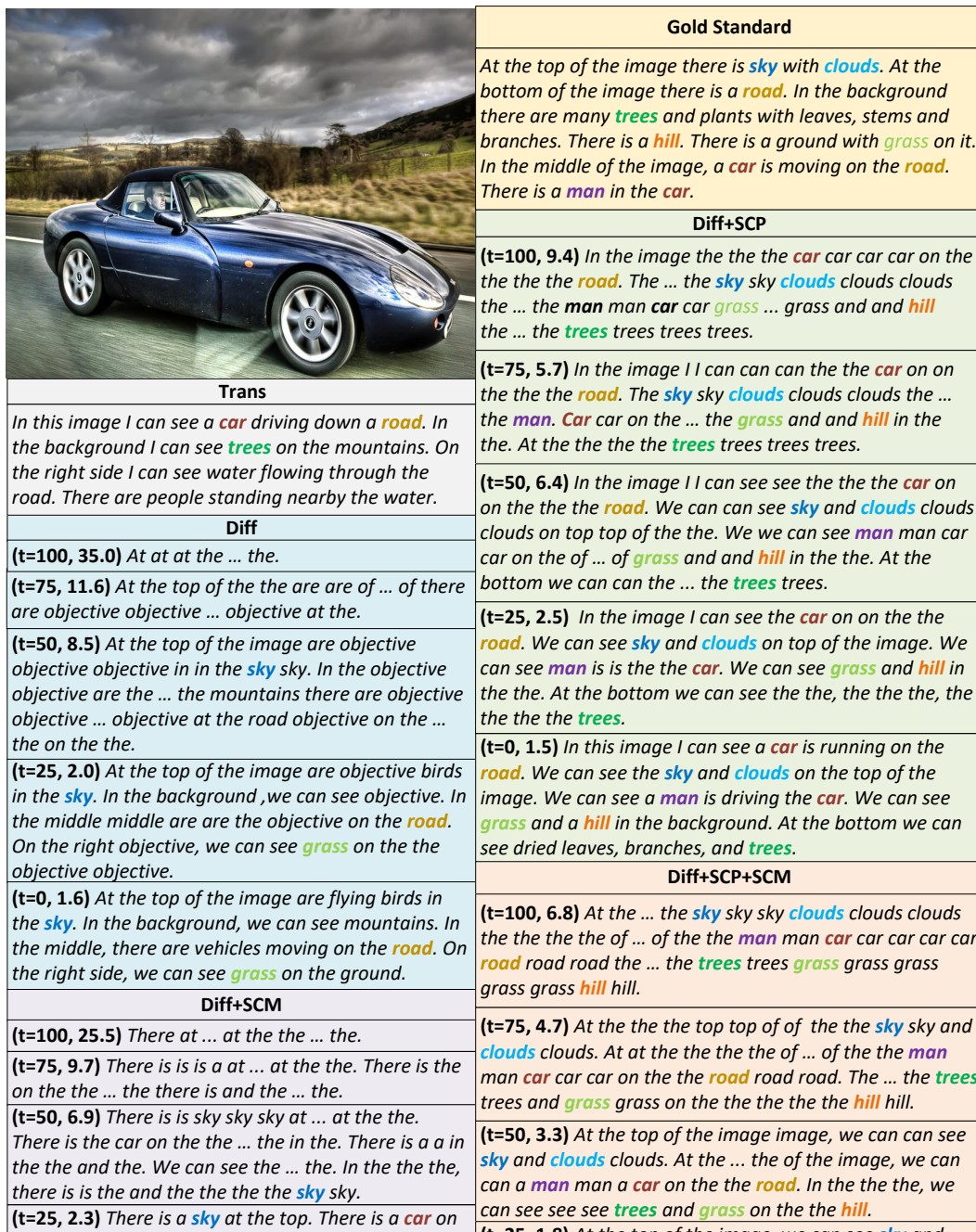

Figure 7: Descriptions generated by baseline models and our full model with an example input image from LN, where the gold standard is also presented for reference.

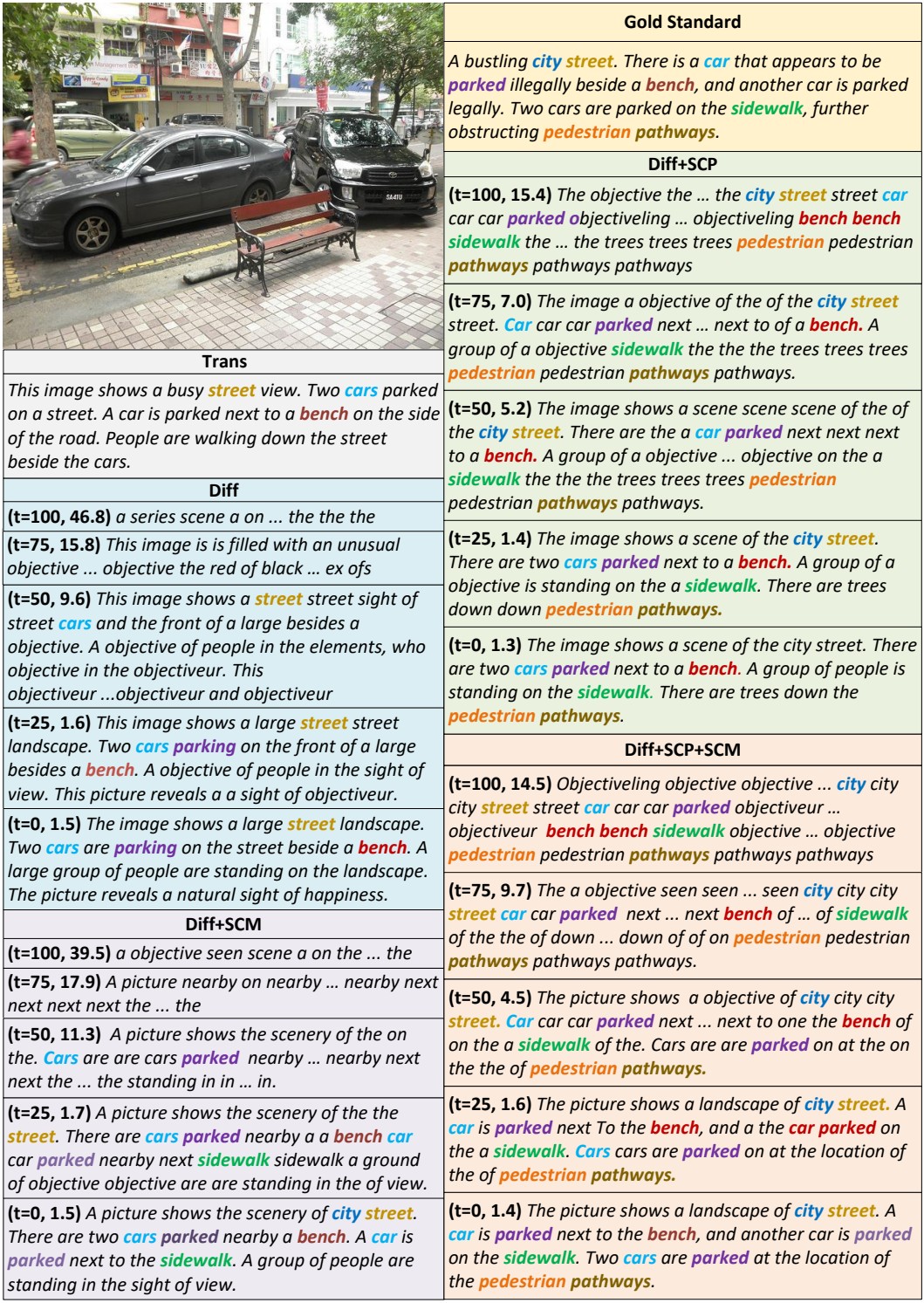

Figure 8: Descriptions generated by baseline models and our full model with an example input image from COCO-LT, where the gold standard is also presented for reference.

