# OpenReview forum: "Semantic Memory Guided Diffusion Networks for Image-to-Long Text Generation"
_ICLR.cc/2024/Conference — Submitted to ICLR 2024_

### Official Review · Reviewer_52FQ · 2023-10-31

**Soundness:** 3 good
**Presentation:** 2 fair
**Contribution:** 3 good
**Rating:** 5
**Confidence:** 3

**Summary:**

This paper introduces a diffusion-based text generation method I2LTG which uses semantic concepts extracted from training datasets to help with diffusion-based text generation. Semantic Concept Predictor predicts the relevant semantic concepts given initial semantic matrix and visual features. Then, Semantic Conditional Memory aggregates the semantic concepts through memory vectors and mechanism before feeding the memory responses to the diffusion decoder. Experiments show that the I2LTG model is able to achieve superior long-text generation performance, compared to existing works.

**Strengths:**

- Predicting semantic concepts as an intermediate representation for (long) text generation is an interesting and unique approach. It also makes good sense intuitively.
- Results are good.

**Weaknesses:**

- The proposed method requires semantic concepts to be obtained from existing datasets. There is no indication that the validation/test set was not used It would be cheating if the method indeed used validation/test set for extracting semantic concepts. There would be strong hints provided to the model.
- Not sure which visual backbone is used. Is it comparable to backbones used by other methods?

**Questions:**

- In 2.3, the word "conditional" is not mentioned at all. What is the point of calling the corresponding component as Semantic "Conditional" Memory?

**Details Of Ethics Concerns:**

There might be information leaking from validation/test set to the extracted semantic concepts (Weakness 1).

---

> ### Author Response · Authors · 2023-11-21
> **Response**
>
> Dear reviwer 52FQ,
>
> Thank you for your valuable comments and suggestions, which will help make our article clearer. We summarize your main concerns as follows.
>
> > The proposed method requires semantic concepts to be obtained from existing datasets. There is no indication that the validation/test set was not used It would be cheating if the method indeed used validation/test set for extracting semantic concepts. There would be strong hints provided to the model.
>
> We would like to clarify the details of extracting the semantic concepts. For the training of SCP, we obtain the semantic concepts based on the following procedures. First, we employ the NLTK POS tagger to annotate POS tags for each word in all image descriptions. Then, we set a threshold to filter out infrequent words. Based on the aforementioned processes, we are able to obtain three semantic concept lists corresponding to the training, validation, and test set. **For the concept list of training set, we manually delete the semantic word that appear in the lists of the valiadtion/test set, where we use the resulted list to train SCP.** Therefore, we make sure that there is no unfair comparison in our experiments. We will make this clear in the final version of our paper.
>
> > Not sure which visual backbone is used. Is it comparable to backbones used by other methods?
>
> For your concerns of **vision backbones**, our approach uses **ResNet-101** as visual feature extractor, which is the same as other approaches (e.g., KiUT, Organ, R2GenRL) on MIMIC-CXR. For the compared approachs on **CC-SBU, LN, COCO-LT**, the image captioning approaches use **Fast R-CNN with initialization of ResNet's model weights, while LLM-based approaches adopt vision Transformer (ViT)**, where the compared approaches adopt stronger vision backbone models than ours. We will expain it clearer in our the final version of paper.
>
> > In 2.3, the word "conditional" is not mentioned at all. What is the point of calling the corresponding component as Semantic "Conditional" Memory?
>
> We call the memory module as "semantic conditional memory", since it enhance the semantic representations from SCP with its stored information, and **provides conditional guidance for the diffusion networks**. We will make this clear in the final version of our paper.

---

### Official Review · Reviewer_Hjjf · 2023-11-01

**Soundness:** 3 good
**Presentation:** 2 fair
**Contribution:** 3 good
**Rating:** 5
**Confidence:** 3

**Summary:**

In this paper, the authors introduce SEMDIFF, a diffusion-based model equipped with memory networks tailored for I2LTG. Initially, SEMDIFF identifies main semantic concepts within images. It then leverages a memory network to convert these concepts into the diffusion networks to seamlessly integrate them, enhancing the long-text generation process. By doing so, SEMDIFF effectively tackles challenges like incoherence in non-AR text generation, particularly evident in lengthy texts by embedding external guidance within the diffusion iterative generation. The experimental evaluations are conducted on three public datasets, along with COCO-LT, which demonstrate the effectiveness of SEMDIFF over existing state-of-the-art solutions.

**Strengths:**

- The paper is overall organized, facilitating a smooth reading experience. Additionally, the inclusion of model overviews and illustrative figures for the components simplifies the understanding of the proposed method.

- The experimental results well validate the approach. Notably, the paper not only presents comparisons with SoTAs, but also comprehensively study the effects of the modules in the framework with ablation studies.

- The authors have included detailed hyperparameter settings in their implementation to enhance the reproducibility.

**Weaknesses:**

- While the paper provides valuable insights, there are areas in the methodological presentation that could benefit from further rigor. Specifically:
    - The definition of $\mathbf{n}$ in Eq (12) is absent from the document.
    - The cross-entropy loss, denoted as $\mathcal{L}_\text{CE}$, is not formally introduced.

While the authors might perceive some of these notations and concepts as commonly understood within the field, it would enhance the clarity and comprehensiveness of the paper to formally define them. Furthermore, this would enhance the presentation clarity and explain these losses relate outputs from various modules, offering readers a more cohesive understanding of the methodology.


- While the authors have illustrated the effects of diffusion decoding across different timesteps through experiments, the results primarily lean towards a qualitative nature. For a comprehensive understanding, would it be feasible to provide quantitative assessments to delineate these differences?

- Some properties of diffusion models are not studied in the paper. For example, the guidance of diffusion models are known for enhancing the correlation between the generation and the semantic condition for better controllability. Does this properties further enhance in the case of long-text generation?

- The diffusion models are also known for its computation complexity in the generation, as it requires thousands of NFE in the generation. How does it increase the model computation compared with baselines is not fully studied in the paper.

- The configuration regarding the settings of diffusion models is not clearly presented in the paper. How do you choose the diffusion schedule? Do you take DDIM/ODE or DDPM/SDE steps in the reverse process?

**Questions:**

Please refer to the Weakness.

---

> ### Author Response · Authors · 2023-11-21
> **Response**
>
> Dear reviwer Hjjf,
>
> Thank you for your valuable comments and suggestions, which will help make our article clearer. We summarize your main concerns as follows.
>
> > The definition of in Eq (12) is absent from the document. The cross-entropy loss, denoted as, is not formally introduced.
>
> We follow the standard noising process of **denoising diffusion probabilistic model (DDPM)** in Eq (12). We will make this clearer and introduce the cross-entropy loss formally in the final version of this paper.
>
> > While the authors might perceive some of these notations and concepts as commonly understood within the field, it would enhance the clarity and comprehensiveness of the paper to formally define them. Furthermore, this would
> enhance the presentation clarity and explain these losses relate outputs from various modules, offering readers a more cohesive understanding of the methodology.
>
> Thank you for pointing this out. We will make all notations and concepts clarified in the final version of our paper.
>
> > While the authors have illustrated the effects of diffusion decoding across different timesteps through experiments, the results primarily lean towards a qualitative nature.
>
> For your concerns of performance under different timesteps varying from T=100 to T=1000, we report the quantitative results on COCO-LT as follows.
>
> |  Time Step   | BLEU-4  |
> |  ----  | ----  |
> | 100  | 0.093 |
> | 200  | 0.094 |
> | 300  | 0.094 |
> | 400  | 0.095 |
> | 500  | 0.096 |
> | 600  | 0.096 |
> | 700  | 0.097 |
> | 40  | 0.098 |
> | 45  | 0.098 |
> | 50  | **0.098** |
>
> It is observed that **there is a slight improvement as T increases from T=100 to T=1000**, since diffusion networks are able to perform a more refined generation process through more iterations. Notably, **the greater T also requires more inference time to generate an entire long text description**, where we find that generating an image description with **around 100 words takes 5 seconds** given T=1000. We will add this experiments in the final version of our paper.
>
> > Some properties of diffusion models are not studied in the paper. For example, the guidance of diffusion models are known for enhancing the correlation between the generation and the semantic condition for better controllability.
>
> In this work, we follow the standard setup of Bit Diffusion, which **does not use any diffusion guidance approach**. Nonetheless, we try improving the image-to-long text generation task with diffusion guidance after the paper submission, and **have already achieved promising results**, which will be released in later studies.
>
> > The diffusion models are also known for its computation complexity in the generation, as it requires thousands of NFE in the generation. How does it increase the model computation compared with baselines is not fully studied in the paper.
>
> We agree with you that the computation complexity problem is existed in the image generation task, where diffusion models usually require thousands of NFE to generate images. **However, in the text modality that exists in the form of tokens, we draw the opposite conclusion.** The computational complexity of autoregressive (AR) models grows quadratically as sentences become longer, **which is particularly significant in the image-to-long text generation task.** Due to the characteristic of parallel generation of all tokens, the computational complexity of diffusion models is less than that of AR models. We will make this clearer in the final version of our paper.
>
> > The configuration regarding the settings of diffusion models is not clearly presented in the paper. How do you choose the diffusion schedule? Do you take DDIM/ODE or DDPM/SDE steps in the reverse process?
>
> We follow the standard setup of diffusion schedule and diffusion sampler in Bit Diffusion [1], which use **cosine noise schedule** and **DDIM sampler** by default. We will add this details in the final version of our paper.
>
> Reference: [1] Chen, Ting, Ruixiang Zhang, and Geoffrey Hinton. "Analog Bits: Generating Discrete Data using Diffusion Models with Self-conditioning." arXiv preprint arXiv:2208.04202 (2022).

---

### Official Review · Reviewer_FnS9 · 2023-11-01

**Soundness:** 3 good
**Presentation:** 2 fair
**Contribution:** 2 fair
**Rating:** 5
**Confidence:** 2

**Summary:**

The paper targets the long image captioning generation. In this paper, a Semantic Concept Predictor is proposed to predict the key concepts in the text, and a set of memories is introduced to enhance the concept representations. The method is tested on different long captioning datasets and the results are promising.

**Strengths:**

The paper proposes an interesting structure to catch the key concepts and enhance them with auxiliary memories.
The techniques are sound and easy to follow.

**Weaknesses:**

I think Experiments need to be improved. Some part of it is confusing. Table 2, shows the contribution of different components. We can find that the simple basic model ("Trans") provides a strong baseline, especially in LN, COCO-LT, and CC-SBU. This erodes the contribution of the proposed methods: As a simple baseline can achieve impressive performance, can the methods developed for MIMIC-CXR also be efficient on them? The authors did not explore this in the experiments. Also, some LLM are evaluated on them, while it is not sure if they are finetuned on the training sets. As a simple 6-layer transformer encoder-decoder can achieve 0.054 of BL4, it is confusing why the LLM like LLAVA achieves only 0.06 after finetuning. I think this is an interesting point the author needs to explore further in the paper.

Checking Table 3, we can find the proposed method beats the SoTA methods in quite limited scales, like 0.412 vs 0.407 for B1, and 0.129 vs. 0.126 for B4. It is hard to tell if the proposed method is really better than the existing ones. One option here to prove their effectiveness is to adapt the proposed modules to the SoTA method.

Section 4.2 investigates 3 different hyperparameters of the methods, while the size of the semantic concept set is also significant to be explored. Answering some questions like the following can make readers understand the method better: Is the semantic concept set the larger the better? Are the predicted concepts the more the better for the long captioning generation?

**Questions:**

Please refer to the weakness.

I am also confused about Fig. 15 and 16. It seems h^hat_0 is a probability of h^hat_1, h^hat_2, ... with other variables. While h is usually used as the "hidden state". I am not sure why h^hat_0 equals the products of a set of probabilities. it seems p(h^hat_0|h^s, h) would be more reasonable here. Please correct me if I have some misunderstanding.

---

> ### Author Response · Authors · 2023-11-21
> **Response**
>
> Dear reviwer FnS9,
>
> Thank you for your valuable comments and suggestions, which will help make our article clearer. We summarize your main concerns as follows.
>
> > ...can the methods developed for MIMIC-CXR also be efficient on them? The authors did not explore this in the experiments. Also, some LLM are evaluated on them, while it is not sure if they are finetuned on the training sets.
>
> In our preliminary experiments, we have actually tried adapting the methods developed for MIMIC-CXR to the image-to-long text generation task, which only obtains around 0.7 BLEU-4 score on COCO-LT dataset. The reason behind is the **domain gap between natural images and medical X-ray images**, where X-ray images differ greatly from natural images from both structures and contents.
>
> > Also, some LLM are evaluated on them, while it is not sure if they are finetuned on the training sets. As a simple 6-layer transformer encoder-decoder can achieve 0.054 of BL4, it is confusing why the LLM like LLAVA achieves only 0.06 after finetuning. I think this is an interesting point the author needs to explore further in the paper.
>
> In our experiments, we evaluate LLAVA **without fine-tuning**. For the explanation of similar performance of ``Trans'' to LLM-based approach, we find that **LLM-based approaches have possibilities in generating irrelevent texts in the produced image description**, thereby obtaining inferior performance on image-to-long text generation datasets.
>
> > It is hard to tell if the proposed method is really better than the existing ones. One option here to prove their effectiveness is to adapt the proposed modules to the SoTA method.
>
> We find that **existing studies have already obtained high performance on MIMIC-CXR**, so that the performance gain of our approach over them is limited. Thus, we further evaluate our approach on **LN, CC-SBU, and COCO-LT** , where our approach obtains more significant improvement compared to previous approaches. Specifically on CC-SBU, LN, and COCO-LT, our approach obtains **0.109, 0.092, and 0.093** BLEU-4 scores, respectively.
>
> Please note that the improvement on MIMIC-CXR is assessed (t-test) wth significance (at **p<0.05** level), and meanwhile, such performance gain is similar in all previous RRG studies and ours seems even bigger than existing studies, which adequately proves its effectiveness, not mention that it also shows consistent and considerable improvements on other datasets.
>
> > ... Is the semantic concept set the larger the better? Are the predicted concepts the more the better for the long captioning generation?
>
> We run experiments of performance on COCO-LT with respect to the semantic matrix size, which is presented as follows.
>
> |  Semantic Matrix Size   | BLEU-4  |
> |  ----  | ----  |
> | 5  | 0.032 |
> | 10  | 0.035 |
> | 15  | 0.039 |
> | 20  | 0.045 |
> | 25  | 0.049 |
> | 30  | 0.065 |
> | 35  | 0.081 |
> | 40  | 0.093 |
> | 45  | 0.080 |
> | 50  | 0.069 |
>
> Results indicate that **the semantic matrix has an optimal value**. When the semantic matrix is relatively small, the model gradually obtains better performance as its size enlarges, where the matrix is able to cover more related concepts with more essential semantic information stored with its vectors. Once the optimal value is reached, the model performance starts to degrade when the size keeps enlarging, since larger matrix size does not help in storing extra helpful semantic information.
>
> > I am also confused about Fig. 15 and 16. It seems $\mathbf{h}_1$ is a probability of $\mathbf{h}_1$, $\mathbf{h}_2$, ... with other variables. While h is usually used as the "hidden state". I am not sure why $\mathbf{h}_0$ equals the products of a set of probabilities. it
> seems $p(\mathbf{h}_0|\mathbf{h}^s, \mathbf{h})$ would be more reasonable here. Please correct me if I have some misunderstanding.
>
> We agree with your opinion that h could be improved in writing.$\mathbf{h}_0$ represents **the clean bit representation that is totally de-noised from Gaussian noise**. We use $\mathbf{h}_0$ as the variables since we regard the intermediate representation of diffusion networks as a type of hidden state. We will fix this in the final version of our paper.

---

### Official Review · Reviewer_1tdH · 2023-11-01

**Soundness:** 2 fair
**Presentation:** 2 fair
**Contribution:** 2 fair
**Rating:** 5
**Confidence:** 4

**Summary:**

This paper proposes a new challenging task, image-to-long-text generation. To overcome the limitations of existing approaches such as the inability to generate sufficiently comprehensive
and complete textual content. The authors introduce a semantic memory-guided diffusion network (SeMDiff), which captures the essential semantic information of images through a semantic concept predictor, and enhances the semantic representation through a semantic conditional memory module. Afterward, a diffusion decoder module is employed to generate comprehensive and coherent long texts.

**Strengths:**

1. The problem of long-text generation this paper addressed is an interesting and important task.
2. The proposed method looks technical and sound.
3. The proposed new dataset COCO-LT is technically reasonable and maybe useful in the future.
4. The paper is well-organized and easy to read.

**Weaknesses:**

1. In section 2.2, the initialized matrix contains a series of semantic vectors to cover all possible concepts, but how to get the semantic vectors of these concepts is not mentioned.
2. The statement of the semantic conditional memory is not clear. In section 2.3, the description of “the memory stores the information in aligning image and texts” is ambitious. What is the specific information here, and how it is obtained?
3. Although the purpose of the proposed approach is to solve the long text generation problem, I think it is still necessary to test on some short caption benchmarks, such as MS-COCO.
4. The metric CIDEr is missed, which is a very important metric in the image captioning task.
5. I think the comparison in Table 2, 3 is not fair. Only SEMDIFF is transformer-based while others are all ResNet-101-based. There have been some other Transformer-based methods [1,2] for image captioning task are proposed. Among them, [1] is also a diffusion-based method. I think comparing with these approaches will further strengthen this paper.

Ref:
[1] Jianjie Luo et al. “Semantic-Conditional Diffusion Networks for Image Captioning”. CVPR, 2023.
[2] Chia-Wen Kuo et al. “HAAV: Hierarchical Aggregation of Augmented Views for Image Captioning”. CVPR, 2023.

**Questions:**

1. In Table 4, the results of existing state-of-the-art solutions reported are zero-shot or fine-tuned, and SeMDiff is the zero-shot or fine-tuned? If the results of SOTA are zero-shot, how about the fine-tuned performance on these datasets?
2. The initialized matrix contains a series of semantic vectors of possible concepts, is it a fixed matrix in the whole training process? Or it will be different for different samples?

---

> ### Author Response · Authors · 2023-11-21
> **Response**
>
> Dear reviwer 1tdH,
>
> Thank you for your valuable comments and suggestions, which will help make our paper clearer. We answer your main concerns as follows.
> > In section 2.2, the initialized matrix contains a series of semantic vectors to cover all possible concepts, but how to get the semantic vectors of these concepts if not mentioned.
>
> In our preliminary experiments, we **randomly initialize** the semantic matrix, where semantic vectors in the semantic matrix are optimized during training to help the concept prediction process. We will explain this more clearer in the final version of paper.
> >The statement of the semantic conditional memory is not clear. In section 2.3, the description of “the memory stores the information in aligning image and texts” is ambitious.
>
> The memory networks **store a series of memory vectors that contain the image-text matching information between the input image and its long text description.** Specifically, we use the stored information through retrieving all memory vectors with the semantic representation (i.e., $\mathbf{h}^s$ computed by Eq. 5), and select the top-$\mathcal{K}$ most related vectors to compute the weighted sum of them to enhance the semantic representation, so that the memory-enhanced representation is able to provide guidance for the diffusion networks to generate the image description with alignment to the input image. To obtain such information, we **randomly initialize** the memory networks at the beginning of the training process, and the memory networks **learn to match the input image and output description until the training is convergence**. We will explain it more clearer in the final version of paper.
> >Although the purpose of the proposed approach is to solve the long text generation problem, I think it is necessary to test on some short caption benchmarks, such as MS-COCO.
> There have been some other Transformer-based methods [1,2] for image captioning task are proposed. Among them, [1] is also a diffusion-based method... The metric CIDEr is missed, which is a very important metric in the image captioning task.
>
> As you mentioned, it would further enhance our reseach if we **evaluate our approach on MS-COCO and compare our approach with SCD-Net and HAAV**. Therefore, we run experiments to compare our performance with them on MS-COCO, and illustrate the results (including CIDEr) in the following table (the results of HAAV come from their paper). We will add the resuts in our the final version of the paper.
>
> |  Model   | BLEU-1 | BLEU-2 | BLEU-3 | BLEU-4 | METEOR | ROUGE-L | CIDEr |
> |  ----  | ----  |  ----  | ----  |  ----  | ----  |  ----  | ----  |
> | HAAV | - | - | - | 0.410 | 0.301 | - | 0.142 |
> | SCD-Net | 0.790 | 0.634 | 0.491 | 0.373 | 0.281 | 0.581 | 0.118 |
> | SeMDiff | **0.795** | **0.651** | **0.503** | **0.412** | **0.308** | **0.598** | **0.145** |
>
> Our approach obtains better performance than HAAV and SCD-Net. Specifically, **for SCD-Net**, it **uses semantic guidance to enhance the diffusion networks** through retrieving related sentence for the generation process.  **Our approach outperforms them with semantic concepts** for description generation, thereby providing more fine-grained guidance to generate elaborated description. **For HAAV**, it adopts multiple encoders to extract multi-view features from images, so as to integrate these features and provide more informative visual information for description generation. By comparing the integrated features with our semantic concepts, **semantic concepts are able to offer more fine-grained guidance for description generation than the integrated visual features from different views**.
>
> >I think the comparison in Table 2, 3 is not fair. Only SEMDIFF is transformer-based while others are all ResNet-101-based.
>
> For your concerns of vision backbones, our approach uses ResNet-101, which is **the same as** other approaches (e.g., KiUT, Organ, R2GenRL) on MIMIC-CXR. Therefore, the comparison is fair. For the compared approachs on CC-SBU, LN, COCO-LT, the image captioning approaches use **Fast R-CNN with initialization of ResNet's model weights**, while **LLM-based approaches adopt vision Transformer (ViT)**, where the compared approaches adopt more powerful vision backbone models than ours. We will expain it clearer in our the final version of paper.
>
> >In Table 4, the results of existing state-of-the-art solutions reported are zero-shot or fine-tuned, and SeMDiff is the zero-shot or fine-tuned? If the results of SOTA are zero-shot, how about the fine-tuned performance on these datasets?
>
> In our experiments, the reported results of existing state-of-the-art solutions are evaluated in a **zero-shot** manner while SeMDiff is **trained from scratch**. We will expain it clearer in our the final version of paper.
>
> For the fine-tuned performance of existing state-of-the-art approaches, SCD-Net (fine-tuned) is able to obtain 0.0008 BLEU-4 score, which is limited compared to our performance (0.0934).

---

### Meta-Review · Area_Chair_oNKc · 2023-12-23

**Metareview:**

This paper proposes a method for generating a long image caption (rather than a short), given an image. The method uses semantic concepts as an intermediate representation, predicting the semantic concepts and using them to to guide a diffusion decoder to generate a caption. The reviewers agree that this architecture makes sense and praised the paper as well-organized and presented. The reviewers did have a number of concerns, most notably concerns regarding the quantitative comparison (1tdH, FnS9, Hjjf) and some notes about presentation (Hjjf, 52FQ). While the authors submitted thoughtful rebuttal, the reviewers were not convinced enough to change the rating. The reviewers are of consensus that the paper is below the bar for ICLR publication. With the concerns above and no champion for the paper, the AC agrees with the reviewer consensus.

**Justification For Why Not Higher Score:**

The reviewers are of consensus that the paper is below the bar for ICLR publication. With the concerns above and no champion for the paper, the AC agrees with the reviewer consensus.

**Justification For Why Not Lower Score:**

N/A

---

### Decision · Program_Chairs · 2024-01-16

Reject